# A simulation-based analysis of the impact of rhetorical citations in science

Honglin Bao [1] ✉ & Misha Teplitskiy [2] ✉

Authors of scientific papers are usually encouraged to cite works that meaningfully influenced their research (substantive citations) and avoid citing works that had no meaningful influence (rhetorical citations). Rhetorical citations are assumed to degrade incentives for good work and benefit prominent papers and researchers. Here, we explore if rhetorical citations have some plausibly positive effects for science and disproportionately benefit the less prominent papers and researchers. We developed a set of agent-based models where agents can cite substantively and rhetorically. Agents first choose papers to read based on their expected quality, become influenced by those that are sufficiently good, and substantively cite them. Next, agents fill any remaining slots in their reference lists with rhetorical citations that support their narrative, regardless of whether they were actually influential. We then turned agents' ability to cite rhetorically on-and-off to measure its effects. Enabling rhetorical citing increased the correlation between paper quality and citations, increased citation churn, and reduced citation inequality. This occurred because rhetorical citing redistributed some citations from a stable set of elite-quality papers to a more dynamic set with high-to-moderate quality and high rhetorical value. Increasing the size of reference lists, often seen as an undesirable trend, amplified the effects. Overall, rhetorical citing may help deconcentrate attention and make it easier to displace established ideas.

Citations are widely used in science to measure the impact of papers and researchers. The assumption underlying this evaluative use of citations is that when writing papers researchers generally cite prior work to acknowledge intellectual debts[1,2]. The debts can be of different types, including methodological, theoretical, and empirical[3]. If most citations are indeed of this substantive type, the enormously laborious and subjective task of assessing the impact of research becomes a relatively straightforward and objective one—just count the citations. However, decades of research show that many citations reflect motivations that have little, if anything, to do with acknowledging intellectual debts[4]. According to the social constructivist theory of citing, authors cite works to persuade readers and reviewers, regardless of whether the works influenced the authors[5–8]. We refer to these as rhetorical citations[9]. Rhetorical citations help authors educate and

persuade audiences, and navigate the publishing process in several ways. Some rhetorical citations provide context[10] or differentiate the citer's contributions from prior works by criticizing them[11]. Some rhetorical citations are coerced during peer review[12]. Consistent with these relatively superficial uses of the literature, studies find that in many cases authors misrepresent the claims of the papers they cite[13]. Authors can usually differentiate whether a citation was made to acknowledge intellectual debts or for other purposes[9], supporting the distinction between substantive and rhetorical citations.

Authors need not be indifferent to a paper's quality when considering citing it rhetorically. For example, prominent papers may bolster the citer's claims more than obscure papers[14], a phenomenon sometimes called "persuasion by naming-dropping"[15,16]. Nevertheless, quality plays at most an indirect role in rhetorical citing—the author's

[1]Harvard Business School, Allston, MA 02163, USA. [2]School of Information, University of Michigan, Ann Arbor, MI 48109, USA. ✉e-mail: hbao@hbs.edu; tepl@umich.edu

own perception of a paper's quality is much less relevant than how potential readers perceive its quality, because the higher the latter, the more persuasive the paper.

Despite the usefulness of some rhetorical citations, the practice has a mixed reputation overall, with some suspecting it to corrupt the literature and incentives for future research[17]. For example, the journal *Nature Genetics* has gone as far as to explicitly warn that manuscripts citing rhetorically will be rejected[18]. Even aside from official policies, some argue that rhetorical citing is a signal of low-quality work[19,20].

The view that rhetorical citing is less desirable than substantive citing implicitly compares the current world with rhetorical citing to a counterfactual world without it. Yet a rigorous comparison between the two worlds does not exist but is worth conducting. This is because the counterfactual world with only substantive citing is likely to be one in which attention is concentrated on only the few best papers, and their advantage becomes locked-in over time. We develop this argument from the literature on how researchers read and cite. When searching for papers to read carefully and potentially cite substantively, researchers focus primarily on quality, which they initially infer through status. As the status of a work grows with citations, it is more and more likely to attract such substantive attention. Teplitskiy et al. use surveys of authors to show that highly cited works attract disproportionally more substantive reading and citing and that the relationship between status and substantive attention is likely causal[9]. Corroborating this finding, Hoppe et al., use the timing of references—whether they were added before or during peer review—to infer their substantive or rhetorical nature, respectively. They find that in the biomedical literature, 11.6% of references are inserted during peer review and these rhetorical references are much more likely to be lower cited papers[21]. Such studies suggest that substantive attention is likely to focus on the highest-status works.

When searching for papers to cite rhetorically researchers consider a work's quality and status as well, but also other, non-quality related factors, i.e., papers' rhetorical value. The weighting of non-quality factors redistributes some attention away from the highest-status works and gives other papers a chance to be cited. Recent empirical evidence shows that the number of uncited papers is declining, likely as a result of the steadily increasing length of reference lists, which allows for more rhetorical citations[22]. If works that remain uncited are less often sources of substantive inspiration, then this reduction—coupled with a decrease in inequality of citation distribution that encompasses uncited papers—can largely be ascribed to rhetorical citing. In other words, while the status of a work is likely to increase both rhetorical and substantive citing, empirical work suggests that the driver is weaker for rhetorical citing. Consequently, rhetorical citing may help weaken the feedback loop and make science more dynamic, at least as measured in terms of citations.

The first contribution of this paper is to compare a counterfactual world with substantive citing only to a realistic world with rhetorical and substantive citing, in order to assess how rhetorical citing affects inequality and dynamism. To do so, we developed a behavioral model of the citing process, which is the paper's second contribution. The model combines substantive and rhetorical motivations to cite, along with cognitively realistic search and reading practices. Formulating a comprehensive theory of citing has been a challenge in information and library science for many decades[4,23,24]. The main existing theories—normative and social constructivist—have been criticized as incomplete as stand-alone theories[6,8]. Recent scholarship focuses on synthesizing the theories, for example, the "social systems citation theory"[25]. Our model contributes towards these efforts by integrating diverse citing motivations.

Empirically, comparing scientific communities with and without rhetorical citing is challenging. First, there may not exist any communities without rhetorical citing[9]. Our model addresses this challenge by simulating artificial communities with arbitrary levels of rhetorical citing. This enables us to turn rhetorical citing on and off and measure its effects. Second, even if such communities did exist, classifying citations as substantive or rhetorical is difficult. For classification, one approach uses machine learning with training data from third-party labelers[26,27], but how well such labels correspond to authors' actual motivations is unclear. Teplitskiy et al.[9] use a survey, asking authors why they cited a specific paper. However, surveys may have response biases and are difficult to scale. Hoppe et al. use the timing of when references were added—before or after peer review—as a signal of their substantive or rhetorical nature, which is more scalable but relies on the existence of multiple versions of a paper. Lastly, even if the classification challenge was resolved, causality would be difficult to establish, as it is very unlikely that any two communities that differ in their citing practices do not differ in any other consequential ways. We address these challenges by building into our model two citation types—substantive and rhetorical.

## Stylized facts about reading and citing

To motivate the model we describe several stylized facts about how researchers search for, read, and cite papers. These stylized facts, some of which are rather cynical, are not ones we endorse or seek to normalize. They are simply practices that, to the best of our knowledge, are well supported by empirical studies.

Researchers do not read all potentially relevant research papers but select which ones to read strategically[28,29]. A key criterion for selecting among relevant papers is quality (or fitness[30])— scientists prioritize reading (even if not citing) the best papers[31]. Identifying the best papers to read is challenging since quality can be hard to discern at a glance. Researchers instead use heuristics[32,33]. One commonly used heuristic is a paper's or author's status, which is assumed to proxy quality[34–36]. Citation count is one component of status[9]. Thus, when scientists search for papers to read, they are likely to prioritize the highly cited ones. Heuristic-based selection can cause attention and citations to increasingly concentrate among the highest cited works, due to the feedback loop between current selections and future selections[33]. Researchers may also use other criteria to select what to read, such as familiarity with the authors and recommendations[37], although how frequently such criteria drive decisions is debated[38]. We do not model these criteria but note that this is a simplification of empirical practices.

Search and reading practices are related to citing practices, with the literature on the latter drawing on two main theories, the normative and the social constructivist. The normative theory posits that there is a norm in science to acknowledge intellectual debts and researchers hold themselves to this norm[4,8,39]. To use more colloquial terminology, we call normative citations "substantive". Substantive citing presupposes reading since it is difficult to be substantively influenced by a work without knowing its contents. The styled facts around selection and cumulative advantage in reading described above should also apply to substantive citing. For example, if a researcher tends to select papers with high perceived quality to read, she will tend to only substantively cite (and be influenced by) papers with high perceived quality (not actual quality).

According to the social constructivist theory, authors also cite works to persuade readers, regardless of whether the works had substantive influences on them[5,8]. We refer to this type of citing as "rhetorical." Because rhetorical citing does not require influence, it does not require close (and, at the extreme, any) reading. The non-necessity of reading is supported by several lines of evidence. Studies comparing what citers of papers claim those papers say vs. what they actually say show frequent disagreements and distortions[40–42], with 9.5% of a sample of psychology citations being outright mischaracterizations of the underlying papers[13]. Cases of very specific mistakes in what a paper is taken to claim or in its actual reference

string are difficult to explain except through a lack of careful reading[43]. Lastly, in surveys, authors report citing papers without carefully reading them and/or without being influenced by them[9].

Researchers are likely to select papers for rhetorical citations based on their rhetorical value, which can be affected by time-invariant characteristics like quality[5] and publication outlet, but also time-varying characteristics like citations and the status of the authors[14,36]. A paper's rhetorical value is likely to increase the more it is cited[32]. High citation counts make papers appear to be of higher quality, more significant, more novel, and more generalizable[9,10,44]. The relative rhetorical value of a paper can thus increase (or decrease) over time.

Rhetorical value of a paper is also likely to differ from researcher-to-researcher, based on how well the paper's claims match the researcher's objectives. For example, when two researchers write on a controversial topic and cite rhetorically, a paper that supports one side of the controversy may be rhetorically valuable for one researcher and less valuable for the other[45]. The rhetorical value of a paper is thus likely to vary between researchers, depend on some time-invariant characteristics like quality and fit, and change over time as it accrues status, citations, or becomes obsolete.

## Metrics of scientific community health

Next, we identify compelling measures of the health of a scientific community and establish whether particular citing practices have deleterious effects on health. After surveying the literature we identified three outcomes and the associated measures that are often seen as capturing aspects of health: citation–quality correlation, citation churn (i.e., the replacement of reference list and disruption of the established canon), and citation inequality. We do not take these dimensions as exhaustive (we return to this point in the "Discussion" section), and only claim that they are commonly discussed. For each of the three metrics we develop a research question, based on the intuition that in a world without rhetorical citing, attention and citations would be highly concentrated among the few elite-quality papers. In contrast, because rhetorical citing depends on factors beyond quality, it redistributes some attention and citations to good-but-not-elite papers. Low-quality papers are generally not cited substantively or rhetorically because they are not good or persuasive. We then consider the moderating effects on these characteristics of the reading budget (how many papers researchers read), citing budget (how many references they may put in a paper), and the literature size (the number of papers relevant to a scientist in a period of time).

Metric 1: Citations-quality correlation. Despite long-standing critiques of citations and metrics derived from them, like journal impact factor and *h*-index, as a measure of quality, in practice they are often used as such by administrators and analysts[4]. Perhaps even more important is how researchers and search engines use such metrics. Researchers perceive papers with higher citations as of higher quality and give them a more substantive reading and citing[9]. They also follow citation trails to locate relevant papers and academic search engines are likely to rank highly cited papers better[46]. Consequently, we assume that the higher the correlation between quality and citations, the better for the community. Rhetorical citing can make the correlation stronger. With only substantive citing, the citation distribution becomes in effect bimodal, with high-quality papers receiving all of them and others receiving 0 citations. With rhetorical citing, researchers consider a variety of factors beyond quality and, consequently, citations are more evenly and proportionally distributed across low- to high-quality papers. The more proportional distribution will have a stronger correlation.

Research Question (RQ) 1: How do rhetorical citations impact the citation-quality correlation?

Metric 2: Citation churn. As knowledge in a field evolves, some ideas receive increasing support and, eventually, may become taken-for-granted or "blackboxed"[47]. In a healthy community, if new ideas arise

that are better than the old ones, they should be recognized as such, and the old ones should be displaced. Such disruptive ideas are associated with higher novelty[48] and are predictive of the highest level of recognition in the scientific community, like the Nobel prize[49]. A robust amount of turnover in reference lists across time, which we call citation churn, may thus indicate a healthy evolution of published knowledge. In contrast, if the same set of papers remains the highest cited decade after decade, the community may experience stagnation. Indeed, empirical work suggests that such stagnation is on the rise[50–53]. Rhetorical citing may increase citation churn by reducing lock-in. Different researchers may find different papers rhetorically useful, e.g., supporting their own claims, and not concentrate their attention and citations on only elite-quality papers. The more equal distribution of citations then weakens the feedback loop from current to future citations.

RQ 2: How does rhetorical citing impact the citation churn?

Metric 3: Citation inequality. Scholarship has long found that the distribution of citations is highly skewed[54] and that highly cited canons increasingly attract new citations over time[22,55,56]. This degree of inequality is often described as problematic[55], and possibly indicative of stagnation[50,53]. While the optimal degree of inequality is debated, there is evidence that the realized degree is affected by factors such as technology[51] and even seemingly irrelevant factors like choice architecture[57]. Here, we follow this latter literature in investigating the effect of rhetorical citing on citation inequality, without taking a strong position on what amount is optimal for science. In a world with substantive citing only, citations would be concentrated among the highest-quality papers, and that concentration would increase via the feedback loop of researchers choosing highly cited works to read and citing them yet more. Rhetorical citing may decrease citation inequality because researchers select papers to cite rhetorically based on a variety of idiosyncratic factors like person-specific rhetorical value, not only quality.

RQ 3: How does rhetorical citing impact citation inequality?

The relationships between rhetorical citing and the metrics above may be moderated by several characteristics of a scientific community. First, consider literature size, or the number of papers relevant to a particular researcher. While literature size can change dramatically over time and topic, scientists' cognitive constraints are relatively stable. The stability of cognition implies that the number of papers scientists are capable of reading and being influenced by is also relatively stable. The larger the literature size, the smaller the fraction of papers a scientist will read and cite. Consequently, the larger the literature (while keeping citing budgets and other factors fixed), the more unequal the citation distribution. Relatedly, the more unequal the citation distribution, the lower the quality–citations correlation.

RQ 4: How does the literature size impact the community health metrics?

We also expect the metrics of community health to be affected by researchers' reading and citing budgets—the number of papers they can realistically read, and are expected by specific fields and outlets to cite, respectively. While researchers' reading practices are difficult to measure at scale, the typical number of references in a paper is easily observable and varies widely across fields and time[58]. We expect researchers to know their reading and citing budgets. Citing budgets are of particular interest from a policy perspective, as they can be and are routinely set by publishing outlets. For example, *Nature's* formatting guidelines mention that research articles typically have no more than 50 references (https://www.nature.com/nature/for-authors/formatting-guide). The more a researcher reads—the higher the reading budget—the more likely she is to give attention to non-elite-quality papers. However, if the researcher is limited to a small set of references, the slots will continue to go to papers

that have had a substantive influence, presumably those of elite quality, and the additional papers will not be cited. When the reference list is expanded, for example by allowing rhetorical citations, researchers have more opportunities to populate it with non-elite-quality but rhetorically useful papers. Larger citing budgets, but not reading budgets alone, should thus reduce citation inequality, improve the correlation between citations and quality, and increase citation churn.

RQ 5A: How does the citing budget impact the community health metrics?

RQ 5B: How does the reading budget impact the community health metrics?

To address the aforementioned research questions, we formalize a family of models that vary from the simplest (homogeneous agents) to more complex and realistic (heterogeneous agents). When agents are homogeneous, they perceive the quality and rhetorical values of papers identically, and when they are heterogeneous, a paper may have a higher topical or rhetorical fit and therefore higher value to one agent than another. Our models follow the logic of classic threshold adoption models[59,60], where agents choose a work to cite if it exceeds some person-specific threshold, with one crucial change: we allow for multiple types of adoption.

In our model, agents have their budgets (the maximum number) for reading and citing. In what we call the full model, agents cite substantively and rhetorically. At each time period $t$ an agent $j$ joins the environment. Agent $j$ first ranks all papers by perceived quality $s_{i,j,t}$, which is a function of peper $i$'s actual quality $q_i$, existing citations $c_{i,t}$ at time $t$, and people's bias ($\text{fit}_{i,j}$ and perception errors $\varepsilon_{i,j}$). Then, they read the $m$ highest perceived-quality papers (reading budget $m$), observe the papers' actual quality $q_{i,j}$ (with fit) in their eyes after reading and proceed to the citing stage, which occurs in two steps. First, adhering to academic norms, agents substantively draw inspiration from and cite all sufficiently good papers in their eyes whose quality exceeds the threshold $q_{i,j} > \tau_j$. If there are any remaining slots in the citing budget $n$, agents rank all papers on rhetorical value $r_{i,j,t}$ and populate the slots with those with the highest $r_{i,j,t}$'s. $r_{i,j,t}$ is a

combination of the paper's underlying rhetorical value $r_{i,j}$ (e.g., to what extent the paper supports the author's own claims) and the paper's status premium $s_{i,j,t}$ (to what extent the paper's quality (with fit) and citations help increase the persuasiveness). We allow an agent to select the same paper for a substantive and a rhetorical slot. The details of the functional relationship between different variables of papers and how these variables are initialized and evolved in the eyes of readers are entailed in the "Methods" section and Table 1.

We compare this full model to two null models, where agents cite only substantively, i.e., according to $q_{i,j}$. The two null models differ on how they treat the case where there are insufficient papers of high enough quality to fill the entire citing budget.

- Null-fixed-reference. Null model with fixed citing budget: Agents cite the $n$ papers with highest $q_{i,j}$, even if they are below the threshold.
- Null-fixed-threshold. Null model with fixed threshold: Agents cite only papers with $q_{i,j}$ above the threshold, even if that leaves unfilled slots in the reference list.

Figure 1 illustrates the modeling approach.

To measure the impact of rhetorical citations, we operationalized the metrics of community health in the following way. To measure the correlation between citation counts $c_{i,t}$ and quality $q_i$ at time step $t$ we used the Pearson correlation coefficient. To measure citation churn at time step $t$, we used the number of papers cited in time $t$ that were not cited in time $t-1$. Larger values represent more churn. To measure citation inequality at time step $t$, we utilized the Gini coefficient of citation distribution in time $t$, computed by dividing the area between the equal cumulative distribution of citations and the actual cumulative distribution of citations by the area under the curve of the equal cumulative distribution. Larger values represent more inequality.

In sum, we build a set of agent-based models based on how people read and cite empirically, measure how turning rhetorical citing on and off affects three relatively measurable and much-discussed metrics of community health—the correlation between

## Table 1 | Parameters used in the model

| Parameter name | Symbol | Ranges of values | Notes |
|---|---|---|---|
| Underlying quality | $q_i$ | [0,1] | Distribution: Beta(1,6) Robust to other distributions, see Supplementary Discussion 1.1: value distributions. |
| Threshold | $\tau_j$ | [0,1] | Distribution: Uniform(0,1). Helps determine how many references in the agent's reference list are substantive vs. rhetorical.<br>Robust to the normal distribution, see Supplementary Discussion 1.2: threshold. For homogenous citers, $\tau_j = 0.5$ |
| Perception error | $\varepsilon_{i,j}$ | ≈[−0.15,0.15] | Distribution: Normal(0, 0.05). Min and max values are appx. ±3*SD = ± 0.15.<br>Robust to other distributions, see Supplementary Discussion 1.3: perception error. |
| Fit | $\text{fit}_{i,j}$ | [−0.1,0.1] | Distribution: Uniform(−0.1,0.1).<br>Robust to more/less variant fits and their effects on redistributing attention are much lower than rhetorical citing, see Supplementary Discussion 1.4: fit. Person-specific for heterogeneous citers, identical for homogeneous citers. |
| Perceived quality | $s_{i,j,t}$ | [0,2] | See the section "Methods" (Eq. (2)). The maximum citation premium $\alpha \times c_{i,t} = 1$ (see Supplementary Discussion 1.5: reinforcement strengths). Values of $q_i + \text{fit}_{i,j} + \varepsilon_{i,j}$ that are >1 or <0 are set to either 0 or 1, respectively. |
| Overall rhetorical value | $r_{i,j,t}$ | [0, 1.6] | Methods Eqs. 3 and 4. Composed of an underlying rhetorical value $r_{i,j}$ component (Distribution: Beta(1,6). Person-specific for heterogeneous citers, identical for homogeneous citers) and a component that depends on perceived quality. Max value: $1 + 2*\beta = 1.6$.<br>Robust to other underlying distributions and other values of β, see Supplementary Discussion 1.1: value distributions and Supplementary Discussion 1.5: reinforcement strengths. |
| Effect of $c_{i,t}$ on $s_{i,j,t}$ | $\alpha$ | 0.001 | Robust to other reinforcing strengths, see Supplementary Discussion 1.5: reinforcement strengths. The maximum citation premium is 1000 (max cites) * 0.001 = 1 |
| Effect of $s_{i,j,t}$ on $r_{i,j,t}$ | $\beta$ | 0.3 | Robust to other values of β, see Supplementary Discussion 1.5: reinforcement strengths. The maximum perceived quality premium is 2 (max perceived quality)*0.3 = 0.6. |
| Literature size | $N$ | 200–800 | |
| Reading budget | $m$ | 50–150 | |
| Citing budget | $n$ | 20–100 | |
| Time steps | $t$ | 1000 | |

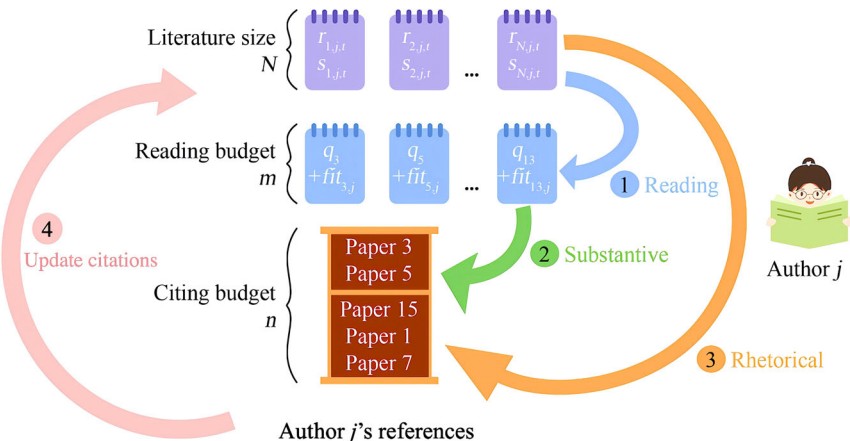

**Fig. 1 | The citing model. Stages are denoted by circled numbers.** For each paper $i$ in the literature ($N$ papers), author $j$ observes its perceived quality $s_{i,j,t}$ and rhetorical value $r_{i,j,t}$ at each time step $t$. Stage 1. An author chooses the $m$ papers with the highest $s_{i,j,t}$'s to read. Reading reveals the papers' quality in people's eyes $q_i + \mathrm{fit}_{i,j}$. Stage 2. An author chooses all papers with sufficient quality and fit for substantive citations, i.e., $q_i + \mathrm{fit}_{i,j} > \tau_j$. Stage 3. If there are any remaining slots in the reference list, the author rhetorically cites papers with the highest rhetorical values $r_{i,j,t}$ until there are no more slots. Finally, the citation count of each paper is updated and timestep is advanced by 1.

citations and quality, citation churn, and citation inequality. We show that rhetorical citations benefit community health by redistributing focus from a small stable set of elite papers to a more dynamic set of high- to mid-quality ones that are rhetorically useful.

## Results

We present results from the more realistic heterogeneous agents models and for completeness report homogeneous agents results in Supplementary Information (Supplementary Methods).

### Effect of rhetorical citing on three metrics of community health

Figure 2 shows the results from the main model specifications. The top row shows how the three metrics of community health—quality-citations correlation (panel a), citation churn (panel b), and citation inequality (panel c)—evolve across the 1000 timesteps. The middle row extracts one summary statistic for each curve in the top row and compares them across models. To test whether rhetorical citations benefit the three metrics of community health, one-tailed OLS regressions were performed without multiple comparisons. Data has been tested to ensure that they meet the assumptions of $t$-test. Panel d shows that, at the end of 1000 timesteps, quality, and citations are more correlated (+2.5%) in the full model than in either of the null models. Full-Null reference: $t(39998) = 120.526$, Cohen's $d = 1.205$, 95% CI = (0.019, 0.019), $n = 20{,}000$ (1000 steps × 20 runs); Full-Null threshold: $t(39998) = 126.128$, Cohen's $d = 1.261$, 95% CI = (0.019, 0.020), $n = 20{,}000$. Panel e shows that churn averaged across the 1000 iterations is 2.36 times higher than the null-fixed-reference ($t(39958) = 484.613$, Cohen's $d = 4.849$, 95% CI = (15.217, 15.340), $n = 19980$ (references in step 2 not in step 1 to references in step 1000 not in step 999; 20 runs)) and 2.17 times higher than the null-fixed-threshold models ($t(39958) = 272.673$, Cohen's $d = 2.728$, 95% CI = (15.103, 15.322), $n = 19980$). Panel f shows that after 1000 iterations, the Gini coefficient, measuring citation inequality, is 30–31% lower than both null models. Full-Null reference: $t(39998) = -1295.954$, Cohen's $d = 12.960$, 95% CI = (−0.292, −0.291), $n = 20{,}000$; Full-Null threshold: $t(39998) = -1308.778$, Cohen's $d = 13.088$, 95% CI = (−0.295, −0.294), $n = 20{,}000$.

To better understand why rhetorical citing affects the community health metrics in this way, we focus on citations to two groups of papers: high-quality (underlying quality $q_i$ in the top 40) and mid-quality papers ($q_i$ in the top 41–150). These groups account for 25% of our literature (600 papers) but attract approximately 85% of the citations. We then measure how many and what type of citations (substantive and rhetorical) the groups get in the full vs. null models. Substantive citing implies that citations should go to the highest-quality papers. Panels h and i show that for the two null models, that is roughly true, with only about 18% of citations going to mid-quality papers. This minority of citations is accrued due to perception errors and variability of $\mathrm{fit}_{i,j}$. In contrast, in the full model (panel g), the fraction of substantive citations going to the highest-quality papers is significantly reduced, and mid-quality papers are more cited, particularly rhetorically. Note that in all models low-quality papers receive very few citations of even the rhetorical kind because their overall rhetorical value is likely low (e.g., low quality/citation counts result in low perceived quality), even if their underlying rhetorical value to an agent is high. Overall, rhetorical citing thus raises the relative visibility of medium-quality papers and results in citations no longer being concentrated on a small, stable set of high-quality papers.

### Moderating effects of citing budget, reading budget, and literature size

To understand the role of the citing budget, we fixed the reading budget at 120 papers, the literature size at 600 papers, and vary the citing budget from 20 to 100. Figure 3 shows how the three metrics of community health change. In panels a and c, at each citing budget the model is run 20 times for 1000 timesteps each, and the metric is calculated at the end of each run. For panel b, the metric is averaged over 1000 timesteps in each run. To test whether citing budget, reading budget, and literature size have any moderating impact on the community health metrics, we conduct two-tailed OLS regressions without multiple comparisons. Increasing the citing budget from 20 to 100 substantially affects all metrics. In panel a, the citation-quality correlation increases by 35.3% (0.68–0.92, $p < 0.001$) in the full model, 31.3% (0.67–0.88, $p < 0.001$) in the null-fixed-reference model, and 35.8% (0.67–0.91, $p < 0.001$) in the null-fixed-threshold model. In panel b, churn increases by 4.59 times (13.12–60.23, $p < 0.001$) in the full model, 2.76 times (6.46–17.80, $p < 0.001$) in the null-fixed-reference model, and 3.83 times (6.47–24.81, $p < 0.001$) in the null-fixed-threshold model. Churn in the full model increases from 2.03 times to 3.38 times higher than the null-fixed-reference model and from 2.03 times to 2.43 times higher than the null-fixed-threshold model. In panel c, citation inequality decreases by 30.0% (0.70–0.49, $p < 0.001$) in the full model, 14.6% (0.96–0.82, $p < 0.001$) in the null-reference model, 14.6% (0.96–0.82,

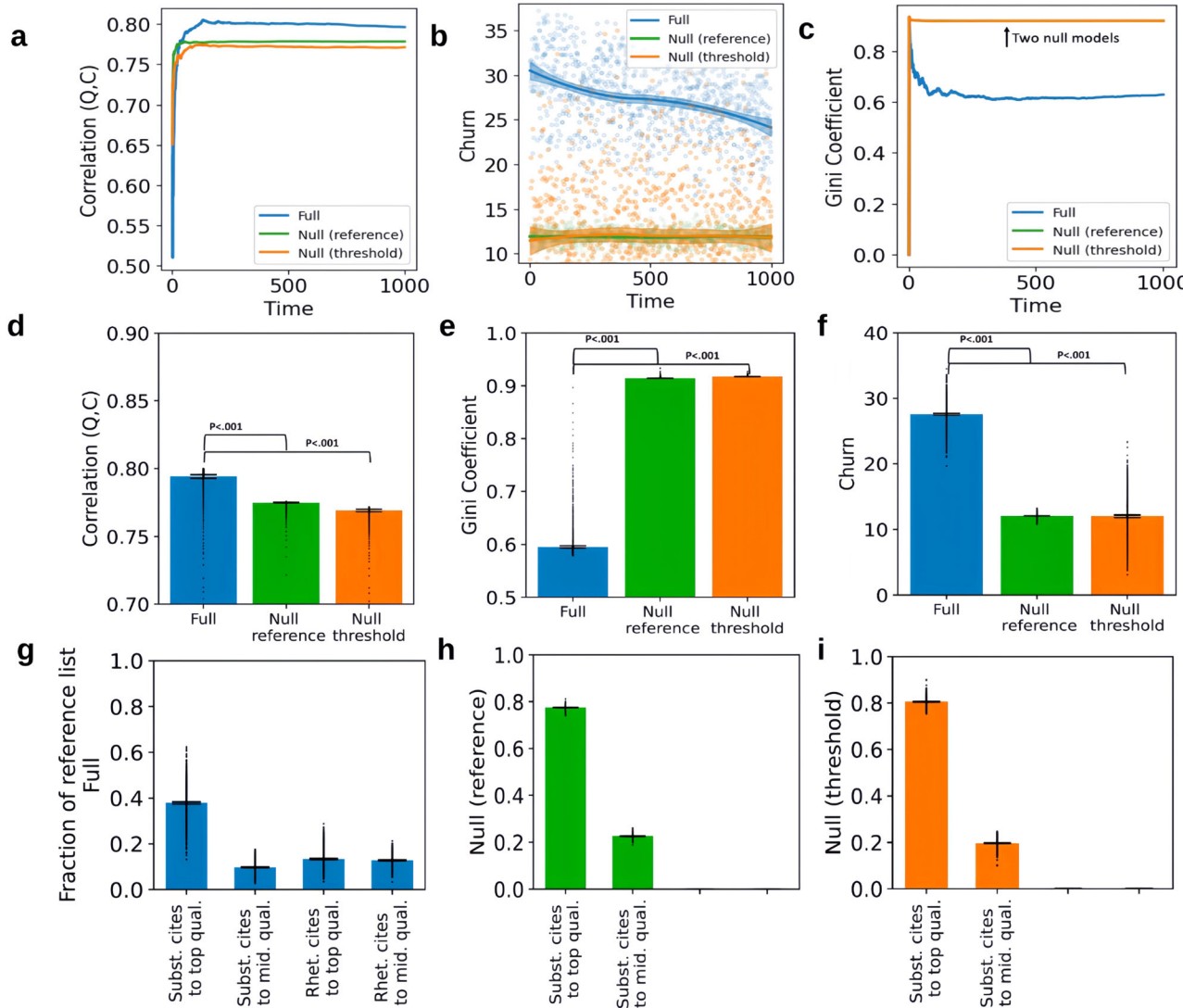

**Fig. 2 | Rhetorical citations benefit community health by redistributing focus from elite papers to a broader set of mid-to-high-quality ones. a–c** how the three metrics of community health evolve across the 1000 iterations in one simulation of the full, null-reference, and null-threshold models. Panel **b** applies locally estimated scatterplot smoothing (LOESS) to elucidate the evolving trend of churn dynamics. Panels **d–f** Rhetorical citations benefit three metrics of community health (averaged over 20 simulation runs). **g–i** The fraction of the reference list (averaged over 20 simulation runs) taken up by substantive citations to papers in the top 40 of quality, substantive citations to papers in the top 41–150 of quality, rhetorical citations to the top 40-quality papers, and rhetorical citations to top 41–150-quality papers. Note: null models (panels **h**, **i**) only have substantive citations. The shaded regions and error bars denote bootstrapped 95% confidence intervals derived from 20 simulation runs. Bars extend from the sample mean as the central line, reaching out to cover a span that is calculated as 1.5 times the interquartile range from the upper and lower quartiles. Outliers have been marked.

$p < 0.001$) in the null-threshold model. Intuitively, increasing the citing budget gives authors more opportunities to cite less elite-quality but still good-quality papers.

To understand the role of the reading budget, we fix the citing budget at 40, the literature size at 600, and vary the reading budget from 50 to 150. Figure 4 shows how the three metrics of community health change across reading budgets. Unlike citing budget, increasing the reading budget has much weaker effects on the metrics. The citation-quality correlation increased very weakly in all models (panel a: full model: 0.79–0.80, $p = 0.010$, coefficient = 1.723e−05; null-fixed-reference model: 0.76–0.78, $p < 0.001$, coefficient = 8.232e−05; null-fixed-threshold model: 0.76–0.78, $p < 0.001$, coefficient = 4.704e−05). Citation churn changed slightly in three models (panel b: full model: 25.51–26.91, $p = 0.005$, coefficient = 0.004; null-fixed-reference model: 8.38–10.89, $p < 0.001$, coefficient = 0.010; null-fixed-threshold mode: 11.38–11.67, $p = 0.938$, coefficient = −7.512e−05). Citation inequality did not change substantially (panel

c: full model: 0.63–0.61, $p = 0.016$, coefficient = −8.85e−05; null-fixed-reference model: 0.92–0.91, $p < 0.001$, coefficient = −2.666e−05; null-fixed-threshold model: 0.93–0.92, $p < 0.001$, coefficient = −1.550e−05). Overall, the reading budget did not have a substantial effect on the three metrics of community health. While it may be epistemically valuable to read more papers, the key constraint on whether those papers get formal recognition in the form of citations is the citing budget.

To understand the role of literature size, we fix the reading budget at 120, the citing budget at 40, and vary the literature size from 200 to 800. Figure 5 shows how the three metrics of community health change across literature sizes. Increasing literature size has substantial but mixed effects on the metrics. When the literature size increases from 200 to 800, the citation-quality correlation decreases by 22.1% in the full model (0.95–0.74, $p < 0.001$), 21.7% in the null-fixed-reference model (0.92–0.72, $p < 0.001$), and 22.6% in the null-fixed-threshold model (0.93–0.72, $p < 0.001$). Citation churn increases by 26.3% in the

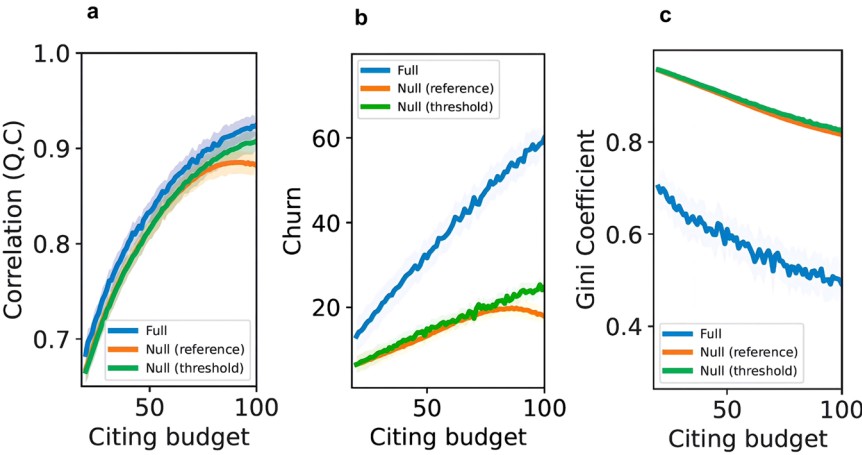

**Fig. 3 | The moderation effect of citing budget.** How increasing the citing budget from 20 to 100 affects the correlation between citations and quality (panel **a**), churn (panel **b**), and citation inequality (panel **c**). For each panel, the shaded areas indicate the bootstrapped 95% confidence intervals derived from 20 simulation runs, while the lines depict the average values across these runs.

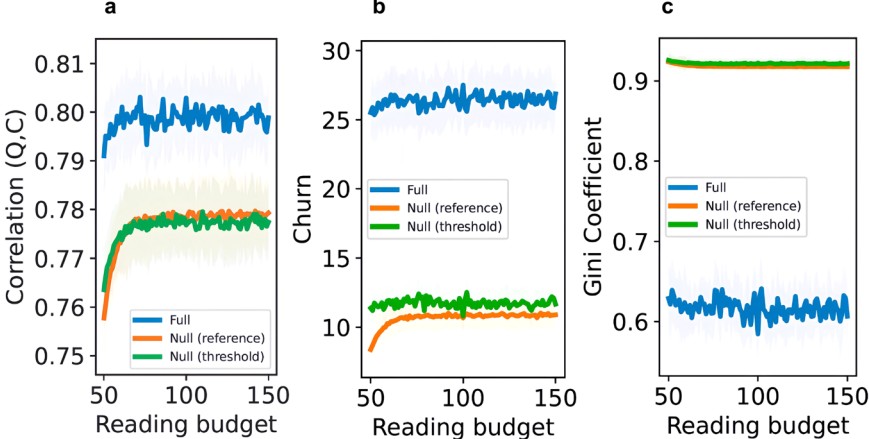

**Fig. 4 | The moderation effect of reading budget.** How increasing the reading budget from 50 to 150 affects the correlation between citation and quality (panel **a**), churn (panel **b**), and citation inequality (panel **c**). For each panel, the shaded areas indicate the bootstrapped 95% confidence intervals derived from 20 simulation runs, while the lines depict the average values across these runs.

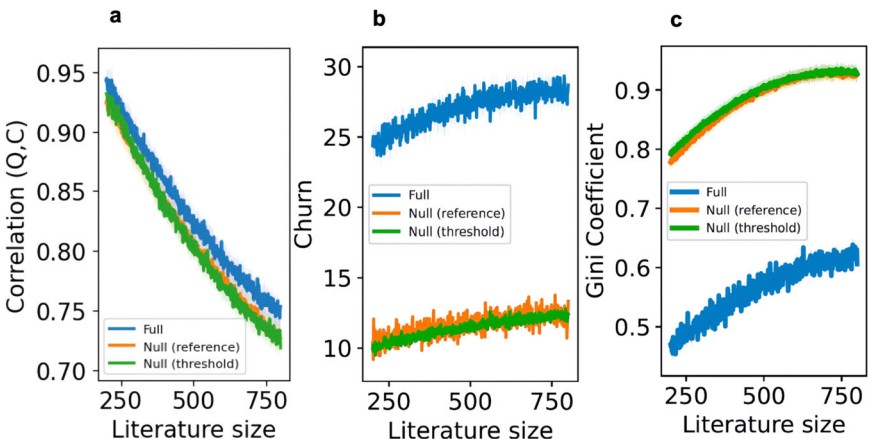

**Fig. 5 | The moderation effect of literature size.** How increasing the literature size from 200 to 800 affects the correlation between citation and quality (panel **a**), churn (panel **b**), and citation inequality (panel **c**). For each panel, the shaded areas indicate the bootstrapped 95% confidence intervals derived from 20 simulation runs, while the lines depict the average values across these runs.

full model (22.52–28.43, $p < 0.001$), 29.3% in the null-fixed-reference model (9.21–11.91, $p < 0.001$), and 22.8% in the null-fixed-threshold model (10.16–12.48, $p < 0.001$). Citation inequality increases by 30.6% in the full model (0.49–0.64, $p < 0.001$), 22.1% in the null-fixed-reference model (0.77–0.94, $p < 0.001$), and 20.5% in the null-fixed threshold model (0.78–0.94, $p < 0.001$). Overall, increasing literature size decreases the correlation between citations and quality, and increases citation churn and inequality. Intuitively, as literature size grows, the fraction of papers read and cited decreases, with both being concentrated on elite-quality pieces. Surprisingly, citation churn increases as well. As the literature size grows, the more high-quality papers (above the threshold) there are. Idiosyncratic factors like $\text{fit}_{i,j}$ can make authors substitute one high-quality paper for another.

## Discussion

Citing papers for reasons other than to acknowledge their influence (rhetorical citing) is generally less encouraged because people believe that it corrupts the literature and incentives for future research. The assumption is intuitive and supported by numerous examples of "undesirable" effects[43,61–63]. It is thus very tempting to envision the counterfactual world with only substantive citing practices optimistically. The first contribution of the paper is to model that counterfactual world realistically, and compare it to the present. Performing the comparison above necessitated the development of a behavioral model of citing, in which researchers are cognitively constrained and cite for both substantive and rhetorical reasons. Such a synthetic model of citing has proven to be an elusive goal for decades of scientometrics and information science literature[4,23], with limited progress to this day. The paper's second contribution is to offer one such synthetic model, drawing on the rich empirical literature on how researchers search for, read, and cite papers. By turning rhetorical citing on and off in the model, we study how it affects three metrics of community health—the correlation between citations and quality, citation churn, and citation inequality. This reveals that rhetorical citing changes, and arguably improves, three aspects of scientific community health. The proximate explanation for these effects is that rhetorical citing redistributes citations from the few elite-quality papers to a more diverse set. The more fundamental explanation is that when seeking papers to cite rhetorically, researchers select on factors beyond just quality (which may still be important), such as rhetorical value. Furthermore, increasing the length of reference lists (citing budgets), usually seen as a problem, increased churn, citation–quality correlation, and decreased citation inequality. Increasing the reading budget, usually encouraged, had little effect on three metrics. Increasing the literature size had mixed effects, increasing churn, decreasing citation–quality correlation, and increasing inequality. While previous work has pointed to the volume of research as a driver of stagnation[50], our work reveals that some seemingly undesirable practices in science, like citing papers without being influenced by them, can help mitigate it. This finding points to a broader conclusion—citations are the outcome of a longer process, driven by how researchers search for and read papers. Consequently, attempts to improve only the last part of this citing process without improving the earlier steps may be of limited utility and may even have unintended consequences.

Models such as ours necessitate many simplifications and scope conditions, which we believe are fruitful directions for future research. First and foremost, the three metrics of community health we focus on are not exhaustive or unambiguous in interpretation. While it is relatively unambiguous that a higher citation–quality correlation is better for science than a lower one, the optimal levels of churn and inequality are more ambiguous, although the current literature raises concerns that the current levels are higher than optimal[50,55]. If one takes the conservative view that only one of our three metrics has an unambiguous interpretation, then a conclusion

of our results is that rhetorical citing has some effects that are plausibly positive. However, other potential metrics, like the amount of low-quality information in the literature[64] or inefficient allocation of rewards, were not included. We hope our work stimulates effort to model more dimensions of community health to better capture the overall effects of a different world.

Second, we assumed that the agents and the types of papers they produce remain fixed across the different worlds. In other words, we did not account for how agents might change their behavior in response to the types of citation practices that exist in a community. For example, in a world with only substantive citing, agents may seek to produce papers of high quality rather than high rhetorical value. Note that while incentives that induce papers of high quality only may appear to be ideal, such a world may have the same concerns as the substantive-citing-only worlds this paper explored.

Third, it is important to acknowledge that our models primarily focus on capturing the dynamics of reading and citing within the short-to-medium term. Over a longer period, one should consider additional dynamics such as the scientific obsolescence of older works, the diminishing weight of previous citations, and the introduction of new research. It is also worthwhile to study the effects of more complex reading practices. For example, authors may group papers into tiers, choosing to read everything they can find if it is in their topic and discipline (the top tier), reading more selectively when it is in the same topic but a different discipline (the next layer), and so on. Additionally, we model the underlying quality of a paper with a single constant, although recent evidence suggests that different types of quality have distinct citation patterns[3], which merits further exploration in future studies.

Fourth, the model depends on a number of parameters and their distributions deserve further exploration. For example, our robustness checks in the Supplementary Information (Supplementary Discussion) suggest that the more rhetorical citing is driven by papers' status (perceived quality), or the more substantive citing is infused with variations, the less distinction there is between them. Introducing these and other features can help align the model even more closely with the complexity of empirical reading and citing practices.

The study did not measure the total effect of rhetorical citing. Such an analysis would require taking into account many more direct and indirect effects, such as misinformation in the literature[13,64] and early vs. late-stage references[21]. Consequently, this paper refrains from endorsing rhetorical citing as beneficial overall, and only argues that it is not a priori obvious whether the scientific community is better off without it. More broadly, this work suggests that when considering policies to fix a particular dysfunction in research, it is important to account for the broad set of incentives and cognitive constraints within which researchers operate.

## Methods
### Parameters

A crucial question is how quality and rhetorical values are distributed. We ground the distributions using expert ratings from peer review. Focusing on the prominent *ICML* conference, the reviewing platform *OpenReview.net* provides ratings of submitted papers. Each submitted paper undergoes a two-stage review process, with two reviewers evaluating it at each stage on originality, technical soundness, clarity, significance, and relevance to the literature[65] (https://icml.cc/). The ratings averaged across a paper's four reviewers follow an approximately normal distribution. No papers received maximum points or 0. Assuming that only papers that score in the top 20-30% are accepted (published) leads to a long-tail distribution of ratings of published papers. Note that because our model depends heavily on papers of the highest value, it is not sensitive to the shape of the distribution for low-quality papers. Similarly, team performance in large-scale real-world data exhibits a long-

tail distribution rather than a normal distribution[66]. We thus use a beta distribution $\beta(1,w)$ for both quality and initial rhetorical value. Parameter $w$ determines the fatness of the distribution's tail, which we set at $w = 6$ as it best fits the distributions of ratings of 2019–2022 submissions to *ICML*. In Supplementary Discussion 1.1, we test the robustness of our results to different choices of distributions and $w$'s and find that they are qualitatively consistent with the main choices.

The empirical evidence on how much researchers read (reading budget) or how many relevant papers there are for a particular project is limited. Tenopir et al. surveyed university scholars in 2012, and they found the mean of monthly article readings was about 20[28]. In contrast, how much researchers cite (citing budget) is readily measurable. According to Lancho-Barrantes et al., the average number of references in papers varies between 20 and 50 across fields[67]. To initialize the models, we set the literature size to 600, the reading budget to 120, citing budget to 40. We explore other parameter choices below and in the Supplementary Information (Supplementary Discussion). All parameters used in our paper are shown in Table 1.

## Dynamics

Here we show the model details of how different variables of papers are interconnected and evolved over time.

- Quality threshold: represented by $\tau_j$ for agent $j$. Agents substantively adopt (cite) a paper when its value to them exceeds a threshold. Homogeneous agents have identical thresholds, while heterogeneous agents differ in their thresholds. For agents with very high thresholds, very few papers will meet that bar for a substantive citation, so more of the reference list will be composed of rhetorical citations. In this way, thresholds help determine the composition of the reference list, even when its overall length is fixed.

- Quality and fit: A paper $i$'s underlying quality is represented by $q_i$ and distributed as $Beta(1,6)$ in the paper population. Underlying quality does not have an index $j$ because it is assumed to be identical for all agents. For simplicity, we do not consider different types of underlying quality, such as methodological or theoretical novelty[3,65]. Fit denotes the substantive usefulness of paper $i$ for agent $j$, represented by $\mathrm{fit}_{i,j}$. Fit is expected to vary across agents due to differences in topic or preferences, i.e., agents may choose not to read even a terrific paper if it is on too unrelated a topic. For homogeneous agents, $\mathrm{fit}_{i,j} = 0$. $\mathrm{fit}_{i,j}$ will raise or lower the quality $q_{i,j}$ of paper $i$ in agent $j$'s eyes as in Eq. (1):

$$q_{i,j} = q_i + \mathrm{fit}_{i,j} \tag{1}$$

- Perception error: represented by $\varepsilon_{i,j}$. It is the error agent $j$ makes in perceiving paper $i$'s quality. While a paper's rhetorical value (see below) and fit are relatively easy to judge from skimming, quality is more difficult to judge and is initially perceived with error. This perception error disappears after an agent reads the paper.

- Perceived quality: represented by $s_{i,j,t}$, it denotes the quality of paper $i$ as perceived by agent $j$ at time $t$, before reading. We assume that the higher a paper's citation count $c_{i,t}$ the higher its perceived quality, with the premium determined by a parameter $\alpha$, i.e., reinforcement strength. Adding all the determinants of a paper's perceived quality yields Eq. 2:

$$s_{i,j,t} = q_i + \mathrm{fit}_{i,j} + \alpha \times c_{i,t} + \varepsilon_{i,j} \tag{2}$$

- Overall rhetorical value: represented by $r_{i,j,t}$, denotes the rhetorical usefulness of paper $i$ for agent $j$ at time $t$. Unlike quality, which has an underlying-quality component $q_i$, rhetorical value is assumed to vary substantially from person to

person. For example, a paper taking a side on a debate might be rhetorically useful to those on the same side but not on the other. For heterogeneous agent $j$, paper $i$ has an initial person-specific rhetorical value $r_{i,j}$. For homogenous agents, $r_{i,j} = r_i$. Perceiving the rhetorical value does not require a careful reading so we include a perception error. We assume that rhetorical value increases with perceived quality $s_{i,j,t}$ as in Eq. (3):

$$r_{i,j,t}|\mathrm{unread} = r_{i,j} + \beta \times s_{i,j,t} \tag{3}$$

The parameter $\beta$ determines the strength of the reinforcement process of $s_{i,j,t}$. Note, if the agent has not read the paper, her $s_{i,j,t}$ is affected by the perception error in the perceived quality, as shown in Eq. (2). If the agent has read the paper closely, the perception error in perceived quality disappears, and the rhetorical value becomes

$$r_{i,j,t}|\mathrm{read} = r_{i,j} + \beta \times (s_{i,j,t} - \varepsilon_{i,j}) \tag{4}$$

A key feature of adoption models like these is the strength of the reinforcement or social influence process—the degree to which an agent's adoption in a time period is determined by adoption by other agents in a previous period[68,69]. We parametrize the strength of reinforcement with two parameters $\alpha$ and $\beta$, for substantive and rhetorical values respectively. Additionally, modelers sometimes include mechanisms that prevent runaway reinforcement, which can lead to the unrealistic phenomenon of guaranteed adoption[68,70]. For example, some assume that the reinforcement effect decays over time[30]. To simplify the model, we do not impose a ceiling on the strength of reinforcement, which is clearly unrealistic in the long-run. Consequently, a key scope condition of our models is that they model citing in the short-to-medium run.

The models are run for 1000 timesteps. At each timestep, one agent makes her citing decisions (i.e., publishes one paper, and thereby cites several from the literature). Then, the citation counts and the quantities that depend on them (perceived quality and rhetorical value) are updated for all papers. For simplicity, the newly published paper is not added to the literature. Thus, any paper from the literature can accrue at most 1000 citations by the end of the run. The model is equivalent if 1000 different agents publish one paper each or one agent publishes 1000 different papers.

## Reporting summary

Further information on research design is available in the Nature Portfolio Reporting Summary linked to this article.

## Data availability

Data generated in this study have been deposited in a persistent GitHub repository https://github.com/Honglin-Bao/rhetorical_citing (https://doi.org/10.5281/zenodo.10038833).

## Code availability

The simulation code for all models has been deposited in a persistent GitHub repository https://github.com/Honglin-Bao/rhetorical_citing (https://doi.org/10.5281/zenodo.10038833).

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

## Acknowledgements

We are grateful to Charles Gomez, Charles Ayoubi, Inna Smirnova, Wei Yang Tham, and seminar participants at the Digital, Data, and Design Institute at Harvard Business School for helpful feedback.

## Author contributions

H.B. and M.T. designed the simulation model. H.B. programmed the model and performed the experiments and analysis. H.B. wrote the initial draft. H.B. and M.T. edited the draft. M.T. supervised the work.

## Competing interests

The authors declare no competing interests.
