## [Peer Review File · Nature Communications]

A simulation-based analysis of the impact of rhetorical citations in scienceReviewers' Comments:

Reviewer #1:

Remarks to the Author:

The authors investigated substantively and rhetorically citing using agent-based models. Whereas rhetorically citing is usually seen as a disfunction of the scientific communication system, the authors argue and demonstrate empirically the important function of rhetorical citing for the system. Rhetorical citing reduces citation inequality and increases the correlation between intrinsic quality and citation counts. The study is remarkably interesting, the used methods are novel (in this context) and seem to be adequate and the analysis leads to interesting results. I recommend publishing the manuscript after a careful revision:

Page 4: The authors assume a definition of rhetorical citing that does not reflect rhetorical citing decisions. The authors argue that rhetorical citing reduces locked-in effects and the "weighting of non-quality factors redistributes some attention away from the highest-status works and gives other papers a chance to be cited." The problem with these arguments is that rhetorical citing is based on factors that are coupled with reputation in science: authors tend to cite reputable authors (institutions) to enhance the meaning of own statements. Research has shown that factors that are not related to scientific quality do not play a key role in rhetorical citing (e.g., <https://doi.org/10.1371/journal.pone.0274810>). Thus, I would argue – other than the authors – that rhetorical citing leads to locked-in effects (key word: Mathew effect).

Page 4/Page 25: On page 4, the authors state that there are only two existing theories on citing without attempts to combine both theories. On page 25, the authors write that "such a synthetic model of citing has proven to be an elusive goal for decades of scientometrics and information science literature (Bornmann & Daniel, 2008; Cronin, 1981), with limited progress to this day". This recent paper (<https://revista.profesionaldelainformacion.com/index.php/EPI/article/view/86939>) shows that new attempts for a citation theory exist.

Page 14: The authors write that "unlike quality, which has an underlying-quality component q_i , rhetorical value is assumed to vary substantially from person to person". The opposite assumption is also possible. We know from (all) studies that have investigated inter-referee agreement in journal peer review that the agreement between two referees assessing the quality of the same manuscript is low. Thus, one can question the existence of an underlying single quality component. Furthermore, one can assume that rhetorical citing is oriented towards reputation which may lead to the reduction of variation between rhetorical values for manuscripts.

Reviewer #2:

Remarks to the Author:

This paper addresses the consequences of "rhetorical" citations, as opposed to "substantive" citations. Rhetorical citations are those made without careful reading or serious influence, with intent to persuade a reader or satisfy reviewers, on citations patterns in a field. The authors develop an agent-based simulation of citation practices in science in which they can tune the amount of rhetorical citation taking place and observe the consequences.

On the positive side, I found the paper to be a very interesting read. The question is an intriguing one, and the issues are important.

I feel that the exposition would be substantially improved by taking care to draw clean lines between the normative/prescriptive arguments and the descriptive treatment of issue of rhetorical citation. While the paper itself takes a modeling approach that would be well-served by descriptive neutrality as to the effects of such processes, the text in many places suggests a normative element. (It is fine, of

course, to have a couple of paragraphs about how this distinction has taken on normative force, as the authors do). We see this even in the title of the paper with reference to 'bad' citations. Example citations notwithstanding, I'm unconvinced that rhetorical citation, broadly construed, is viewed as negatively as the authors suggest. Rather, I think that some elements of rhetorical citation are thought to be best practice. For example, it can be very useful to readers for authors to cite useful review papers in an introduction, even if the authors themselves did not rely on those review papers for their own knowledge of a field. As presently written, the paper seems to be promoting the negative aspect of rhetorical citation in the introductory sections almost as a straw man indented to the finding — that rhetorical citation can be helpful to some measures of field health — seem more surprising.

The model assumes that the importance of author-specific fit is relatively modest in comparison to paper quality. It's not obvious to me that this is the case any many fields. If, for example, I am writing a paper about citation practices in psychology, it seems quite plausible that I'd read *everything* written about citation practices in psychology, regardless of quality, but only some of the literature on citation practices in general. Given this, to what degree is it really fair to say that "In a world with substantive citing only, citations would be concentrated among the highest-quality papers, and that concentration would increase via the feedback loop of researchers choosing highly cited works to read and citing them more"? I would have thought that specificity would already do a fair bit to help out getting authors to read broadly in the literature irrespective of previous citation.

I think there is a much stronger argument for the benefits of having a high correlation than the authors present in section 3 metric 1. Far more important than what administrators and analysts do is what researchers do. Given that researchers (1) follow citation trails and (2) using search engines such as Google Scholar that prioritize highly cited papers, the higher the correlation between citations and quality, the higher the average quality of the papers that get read will be.

For me, the coolest result is that reference list lengths help and help a lot across the board. It's a bit surprising that reading budgets are not nearly as useful.

Were hypotheses 1-4 pre-specified and preregistered, or are they a rhetorical device for the presentation? If the latter, I am uneasy with framing claims as hypotheses unless this is done before the results are known.

My biggest problem with the paper is that I simply don't find it convincing as an argument about what happens in the real world. This is admittedly stylistic preference about models but I find that the present one lands in an uncanny valley between simple analytic models that reveal relationships that are almost unavoidable, and highly detailed models that try to encapsulate much or all of what is known of real-world behavior. Instead we get a complicated agent-based model that has too many moving parts to be able to easily assess external validity — and here I mean not just to the parameter choices in the model, but to the more fundamental decisions that the authors made in designing a model of citing behavior.

As a result I am convinced that agents who behave as the agents in this model behave will exhibit the comparative patterns described in the paper. But I'm not convinced that these patterns hold for actual researchers' actual citation practices, and I don't know what basic findings to take away about those actual citation practices. And thus I'm not sure how valuable the paper is in the end.

Reviewer #3:

Remarks to the Author:

This is an incredibly interesting and completely original piece of research that is likely to influence and advance the discussion of citation theories and the relation between citation counts and research quality.

The analyses are robust, and the relationship between the stated hypotheses, the proposed model, and the reported results is high.

As such, I have no substantial comments on what is IN the paper, this is an exceptional piece of research.

However, there are topics left out that could deserve some additional consideration from the authors. It may not require inclusion in the analysis but should be considered during the initial framing and the discussion of the results. This is the concept of high-level types of citations and the social processes of citing.

As for the types of citations, consider, e.g., the recent paper by Leahey, Lee and Funk (<https://journals.sagepub.com/doi/abs/10.1177/00031224231168074>). While this paper focuses on novelty, the concepts of theory, methods, and results citations are distinct and very present in how articles are cited. Can these types of citations be held equal in discussing inherent paper quality? Are the ICML ratings of theory, methodology, and results papers comparable? This is more of an example of the underlying problem, namely, which kind of quality the ICML ratings measure. This should be discussed further.

The authors address one type of social process for citing behavior: social influence increases adoption. However, their model overlooks other substantial processes. Namely, authors tend to cite people they are familiar with more (cronyism, in the extreme case, higher discoverability in the more lenient interpretation). One could argue that this influences the reading budget of authors and very clearly which articles are in the pool of papers that can go into the reading budget. This is related to the information-seeking aspect of the paper - do all researchers perform correct, systematic literature searches for their research? This is not what the literature suggests, although there may be field differences in this as well. The ever-increasing amount of available articles to read and cite increases the potential for other signals of virtue, i.e., social markers, as argued by Merton. This entire concept needs to be elaborated. And it also brings me to my only (minor) annoyance with the paper: at one point, the authors state that this paper proposes a synthesis of existing citation theories. I strongly suggest omitting this statement, as it oversells the paper; The statement seems to suggest that the research is a new theory, which is overreaching. It is, however, a highly interesting and important part of the puzzle in developing such a theory.

Reviewer 1

1.1. The authors investigated substantively and rhetorically citing using agent-based models. Whereas rhetorically citing is usually seen as a disfunction of the scientific communication system, the authors argue and demonstrate empirically the important function of rhetorical citing for the system. Rhetorical citing reduces citation inequality and increases the correlation between intrinsic quality and citation counts. The study is remarkably interesting, the used methods are novel (in this context) and seem to be adequate and the analysis leads to interesting results. I recommend publishing the manuscript after a careful revision

We appreciate these kind comments and a perceptive summary of the main message.

1.2. The authors assume a definition of rhetorical citing that does not reflect rhetorical citing decisions. The authors argue that rhetorical citing reduces locked-in effects and the “weighting of non-quality factors redistributes some attention away from the highest-status works and gives other papers a chance to be cited.” The problem with these arguments is that rhetorical citing is based on factors that are coupled with reputation in science: authors tend to cite reputable authors (institutions) to enhance the meaning of own statements. Research has shown that factors that are not related to scientific quality do not play a key role in rhetorical citing (e.g., <https://doi.org/10.1371/journal.pone.0274810>). Thus, I would argue – other than the authors – that rhetorical citing leads to locked-in effects (key word: Mathew effect).

It may be worth distinguishing two ideas. First, we share the view that authors tend to cite reputable authors and institutions to enhance the credibility of their own statements. (Accordingly, in our model the probability of rhetorically citing a paper increases the higher status (more citations, higher quality in people’s eyes (quality + fit)) it is. That occurs by the probability depending on perceived quality S_{jt} , which is proportional to citation count and quality in eyes (see equations (3) and (4)). So, we agree that rhetorical citing leads to lock-in **relative to a hypothetical world with no lock-in at all**.

Second, our argument is that rhetorical citing reduces lock-in **relative to that produced by substantive citing**. In other words, both citing types create lock-in, but the one produced by rhetorical citing is *weaker* (since it weighs quality less). We believe it’s quite reasonable to doubt this argument, in particular the idea that rhetorical citing weighs quality or reputation less. Indeed, we did not hold this idea *a priori* and found it counterintuitive. But we have been “forced” to accept it by the empirical data. Specifically, if rhetorical citing is oriented towards reputation more than substantive citing, we should see that the higher reputation (e.g. citations) of a paper, the more of its citations are rhetorical. But there have now been three separate studies using different methods that show the opposite:

First, our earlier work (Teplitskiy et al. 2022) found that the more highly cited (i.e., higher reputation) the paper the more likely its citations are *substantive*. The key figure is pasted below. Note, the research design ensures that the pattern is not confounded by differences in citing authors or paper age or field.

Fig. 5. Black curve with 95% CI shows predicted probability of a reference having major influence (influence ≥ 4), with fixed effects for the citer and discipline, and controls for year of publication, whether the corresponding author or coauthor added the reference, respondent's expertise in the topics of the reference, and whether it was the first paper in the survey.

Second, Hoppe et al., (2023) supports this hypothesis/idea using a different method. The authors write “Previous work showed that highly cited papers are more likely to be the recipients of substantive citations as identified by authors (7). We make a confirmatory observation; the RCR percentile of papers that were cited in the original preprint [i.e. substantive citations] was 2.4% higher than for papers whose reference was added during review [i.e. rhetorical citations] ($P < 0.001$, Wilcoxon Rank Sum test)” (pg 3).

Third, a recent empirical paper by Kozlowski, et al., 2023 may be of interest regarding referencing budgets. They find a decline in the amount of uncited work, likely a result of the trend of longer reference lists. Particularly relevant to our study is that they find a decrease in overall citation inequality, primarily driven by the reduction in uncited works. If works that remain uncited are less often sources of substantive inspiration, then this reduction — coupled with a decrease in inequality of citation distribution that encompasses uncited papers — can largely be ascribed to rhetorical citing (It is also encouraging to see that their empirical results are consistent with our model).

In summary, we believe the empirical evidence is very strong that rhetorical citations go to disproportionately low-status (e.g. cited) works. We do not see how to explain this pattern except that: while reputation is important for getting rhetorical *and* substantive citations, it is even *more* important for getting substantive citations. So when you add a weaker lock-in process (seemingly undesirable) to a stronger one (seemingly desirable), the overall lock-in process is somewhat weakened.

We have revised the Introduction to make this point clearer and add the recent empirical work that appeared after we wrote the initial version of the paper:

Pages 4 and 5. When searching for papers to read carefully and potentially cite substantively, researchers focus primarily on quality, which they initially infer through status. As the status of a work grows with citations, it is more and more likely to attract such substantive attention. Teplitskiy et al. use surveys of authors to show that highly cited works attract disproportionately more substantive reading and citing, and that the relationship between status and substantive attention is likely causal (Teplitskiy et al., 2022). Corroborating this finding, Hoppe et al., use the timing of references – whether they were added before or during peer review – to infer their substantive or rhetorical nature, respectively. They find that in the biomedical literature 11.6% of references are inserted during peer review and these rhetorical references are much more likely to be lower cited papers (Hoppe et al., 2023). Such studies suggest that substantive attention is likely to focus on the highest status works.

When searching for papers to cite rhetorically researchers consider a work's quality and status as well, but also other, non-quality related factors, i.e., papers' rhetorical value. The weighting of non-quality factors redistributes some attention away from the highest-status works and gives other papers a chance to be cited. Recent empirical evidence shows that the number of uncited papers is declining, likely as a result of the steadily increasing length of reference lists, which allows for more rhetorical citations (Kozlowski et al., 2023). If works that remain uncited are less often sources of substantive inspiration, then this reduction — coupled with a decrease in inequality of citation distribution that encompasses uncited papers — can largely be ascribed to rhetorical citing. In other words, while the status of a work is likely to increase both rhetorical and substantive citing, empirical work suggests that the driver is weaker for rhetorical citing. Consequently, rhetorical citing may help weaken the feedback loop and make science more dynamic, at least as measured in terms of citations.

..... The first contribution of this paper is to compare a counterfactual world with substantive citing only to a realistic world with rhetorical and substantive citing, in order to assess how rhetorical citing affects inequality and dynamism.....

1.3. Page 4/Page 25: On page 4, the authors state that there are only two existing theories on citing without attempts to combine both theories. On page 25, the authors write that “such a synthetic model of citing has proven to be an elusive goal for decades of scientometrics and information science literature (Bornmann & Daniel, 2008; Cronin, 1981), with limited progress to this day”. This recent paper (<https://revista.profesionaldelainformacion.com/index.php/EPI/article/view/86939>) shows that new attempts for a citation theory exist.

Thanks for pointing out this theory! We revised the original text accordingly:

Formulating a comprehensive theory of citing has been a challenge in information and library science for many decades (Cronin, 1981; Bornmann & Daniel, 2008; Tahamtan & Bornmann, 2019). The main existing theories – normative and social constructivist – have been criticized as incomplete as stand-alone theories (Cozzens, 1989; Nicolaisen, 2007). Recent scholarship focuses on synthesizing the theories, for example the "social systems citation theory" of Tahamtan & Bornmann (2022). Our model contributes towards these efforts by integrating diverse citing motivations.

1.4. Page 14: The authors write that “unlike quality, which has an underlying-quality component q_i , rhetorical value is assumed to vary substantially from person to person”. The opposite assumption is also possible. We know from (all) studies that have investigated inter-referee agreement in journal peer review that the agreement between two referees assessing the quality of the same manuscript is low. Thus, one can question the existence of an underlying single quality component.

These are great points. Regarding the inter-referee agreement literature undermining the assumption of a single quality component, our model builds that in via the fit_{ij} term – this term makes any one paper’s quality be perceived differently by different people. (Additionally, people’s perception of the paper’s quality also varies due to the perception error term $\varepsilon_{i,j}$ and existing citation counts, although the error term goes away once a person reads the paper). The main specification we chose in the paper has *underlying quality* distributed as Beta (1,6), so the mean of that distribution is $1/7 = 0.14$, and *fit* is distributed Uniform(-0.1, 0.1). So the *fit* can change person-specific *quality* substantially, and if the *fit* distribution gets much fatter then the overall quantity could go negative for many papers. In the Appendix we consider *fit* with higher variance (Uniform(-0.2, 0.2)), which has the effect of muting the differences between full and rhetorical models on Correlation(quality, citations) but retains big differences on the other two community metrics (see Appendix Section 1.4).

We added the following text to the Discussion:

Pg. 26: Lastly, the model depends on a number of parameters and their distributions deserve further exploration..... For example, our robustness checks in the Appendix suggest that.... the more substantive citing is infused with variations, the less distinction there is between them.... Introducing these and other features can help align the model even more closely with the complexity of empirical reading and citing practices.

1.5. Furthermore, one can assume that rhetorical citing is oriented towards reputation which may lead to the reduction of variation between rhetorical values for manuscripts.

This is related to the comment (1.2) above so our response there applies here too. In summary, we believe the comment is intuitive (and we shared this intuition as well) but it is not supported by empirics, which leads to the opposite conclusion.

Additionally, there are two ways in the model that determine how “oriented towards reputation” rhetorical citing is, one direct and one indirect.

- Direct: the parameter β controls the reinforcing strength of *perceived quality* on rhetorical value. In Appendix 1.5 we vary that strength from weaker or stronger than in the Main model. When $\beta = 1$ agents place equal weight on underlying rhetorical value and perceived quality in determining overall rhetorical value. In this regime, the model comes close to the null models without rhetorical citing.
- Indirect: another way in which the model reduces the variation between rhetorical values of papers is in *relative terms* to the variation in perceived quality: when the variation of fit_{ij} is really high, people vary a lot on how they perceive the quality of a paper whereas the variation in underlying rhetorical values remains constant. Appendix 1.4 considers different distributions of fit_{ij} . When the distribution of fit_{ij} is very wide, the full model is very similar to the null models in terms of quality-citation correlation, but the full model still yields more citation churn and less inequality (Figure S9).

Overall, the more that rhetorical citing is oriented towards reputation, or the more variation is introduced in the substantive citing, the less distinction there is between rhetorical and substantive citing. We believe the empirical work “forces” one to conclude that rhetorical citing is less oriented towards reputation.

We added the following text to the Discussion:

Pg. 26: Lastly, the model depends on a number of parameters and their distributions deserve further exploration. For example, our robustness checks in the Appendix suggest that the more rhetorical citing is driven by papers' status (perceived quality), or the more substantive citing is infused with variations, the less distinction there is between them. Introducing these and other

features can help align the model even more closely with the complexity of empirical reading and citing practices.

New References:

- Hoppe, T. A., Arabi, S., & Hutchins, B. I. (2023). Predicting substantive biomedical citations without full text. *Proceedings of the National Academy of Sciences*, *120*(30), e2213697120.
- Kozlowski, D., Andersen, J. P., & Larivière, V. (2023). Uncited articles and their effect on the concentration of citations. *arXiv preprint arXiv:2306.09911*.
- Tahamtan, I., & Bornmann, L. (2022). The social systems citation theory (SSCT): A proposal to use the social systems theory for conceptualizing publications and their citations links. *Profesional de la Información*, *31*(4).
- Kozlowski, D., Andersen, J. P., & Larivière, V. (2023). Uncited articles and their effect on the concentration of citations. *arXiv preprint arXiv:2306.09911*.

Reviewer 2

2.1. This paper addresses the consequences of “rhetorical” citations, as opposed to “substantive” citations. Rhetorical citations are those made without careful reading or serious influence, with intent to persuade a reader or satisfy reviewers, on citations patterns in a field. The authors develop an agent-based simulation of citation practices in science in which they can tune the amount of rhetorical citation taking place and observe the consequences. On the positive side, I found the paper to be a very interesting read. The question is an intriguing one, and the issues are important.

We appreciate the summary and the positives!

2.2. I feel that the exposition would be substantially improved by taking care to draw clean lines between the normative/prescriptive arguments and the descriptive treatment of issue of rhetorical citation. While the paper itself takes a modeling approach that would be well-served by descriptive neutrality as to the effects of such processes, the text in many places suggests a normative element. (It is fine, of course, to have a couple of paragraphs about how this distinction has taken on normative force, as the authors do). We see this even in the title of the paper with reference to ‘bad’ citations. Example citations notwithstanding, I’m unconvinced that rhetorical citation, broadly construed, is viewed as negatively as the authors suggest. Rather, I think that some elements of rhetorical citation are thought to be best practice. For example, it can be very useful to readers for authors to cite useful review papers in an introduction, even if the authors themselves did not rely on those review papers for their own knowledge of a field. As presently written, the paper seems to be promoting the negative aspect of rhetorical citation in the introductory sections almost as a straw man indented to the finding — that rhetorical citation can be helpful to some measures of field health — seem more surprising.

Thanks for this point. We made the following changes to not overstate the extent of how negatively rhetorical citations are viewed:

Title

“The substantive impact of rhetorical citations in science”

Pg 4 Introduction: ...Despite the usefulness of some rhetorical citations, the practice has a mixed reputation overall, with some suspecting it to corrupt the literature and incentives for future research (Penders, 2018). For example, the journal Nature Genetics has gone as far as to explicitly warn that manuscripts citing rhetorically will be rejected (Nature Genetics Editorials, 2017).... The view that rhetorical citing is less desirable than substantive citing implicitly compares the current world with rhetorical citing to a counterfactual world without it....

Pages 26 and 27 conclusions: Citing papers for reasons other than to acknowledge their influence ("rhetorical citing") is discouraged because it is assumed to corrupt the literature and incentives for future research. The assumption is intuitive and supported by numerous examples of "undesirable" effects (Greenberg, 2009; Letrud & Hernes, 2019; Rekdal, 2014; De Vries et al., 2018).

..... The study did not aspire to measure the total effect of rhetorical citing. Such an analysis would require taking into account many more direct and indirect effects, such as misinformation in the literature (Cobb et al., 2023; West & Bergstrom, 2021) and early vs. late-stage references (Hoppe et al., 2023). This paper refrains from making judgments about which citation motivations are inherently more desirable, or from endorsing rhetorical citing as a practice in the real world, but only argues that it is not a priori obvious whether the scientific community is better off without it.

2.3. The model assumes that the importance of author-specific fit is relatively modest in comparison to paper quality. It's not obvious to me that this is the case in many fields. If, for example, I am writing a paper about citation practices in psychology, it seems quite plausible that I'd read **everything** written about citation practices in psychology, regardless of quality, but only some of the literature on citation practices in general. Given this, to what degree is it really fair to say that "In a world with substantive citing only, citations would be concentrated among the highest-quality papers, and that concentration would increase via the feedback loop of researchers choosing highly cited works to read and citing them more"? I would have thought that specificity would already do a fair bit to help out getting authors to read broadly in the literature irrespective of previous citation.

Taking the second part first, it is a very intriguing point. Perhaps one way to phrase it is that researchers may have "tiers" of papers to read. In the top tier, researchers "should" read everything regardless of perceived quality, in the next tier they should read as much as they can and maybe they prioritize perceived quality, etc. We believe adding such a tier-structure is a very worthwhile extension of the model. However, we think it is better left for future work because tiers add complexity, and the model already has many moving parts. We revised the Discussion to add this idea as a limitation of the current model and a direction for future work:

Pg. 26: Authors may group papers into "tiers," choosing to read everything they can find if it is in their topic and discipline (the top tier), reading more selectively when it is in the same topic but a different discipline (the next layer), and so on.

Regarding the importance of *fit* relative to *underlying quality*, we have two thoughts. First, it was difficult to find any first principles from which to decide which of those two quantities

should have a higher variance, and we debated this quite a bit. The main specification we chose has *underlying quality* distributed as Beta (1,6), so the mean of that distribution is $1/7 = 0.14$, and *fit* is distributed Uniform(-0.1, 0.1). So the *fit* can change person-specific *quality* substantially, and if the *fit* distribution gets much fatter then the overall quantity could go negative for many papers. (Note, *perceived* person-specific quality has additional variance coming from the *perception error* and *citation count*). In the Appendix 1.4 we consider *fit* with higher variance (Uniform(-0.2, 0.2)), which has the effect of muting the differences between full and rhetorical models on Correlation(quality, citations) but retains big differences on the other two community metrics. Second, an interpretation of our distribution choices is that they make the model apply to choices in a subset of the literature that's of reasonably high fit to the project, *i.e.*, quite relevant papers.

We added the following text to the Discussion:

Pg. 26: Lastly, the model depends on a number of parameters and their distributions deserve further exploration..... For example, our robustness checks in the Appendix suggest that.... the more substantive citing is infused with variations, the less distinction there is between them.... Introducing these and other features can help align the model even more closely with the complexity of empirical reading and citing practices.

2.4. I think there is a much stronger argument for the benefits of having a high correlation than the authors present in section 3 metric 1. Far more important than what administrators and analysts do is what researchers do. Given that researchers (1) follow citation trails and (2) using search engines such as Google Scholar that prioritize highly cited papers, the higher the correlation between citations and quality, the higher the average quality of the papers that get read will be.

Great point. We revised the relevant section as follows:

Pg 9.: Despite long-standing critiques of citations and metrics derived from them, like journal impact factor and h-index, as a measure of quality, in practice they are often used as such by administrators and analysts (Bornmann & Daniel, 2008). Perhaps even more important is how researchers and search engines use such metrics. Researchers perceive papers with higher citations as of higher quality and give them a more substantive reading and citing (Teplitskiy et al., 2022). They also follow citation trails to locate relevant papers and academic search engines are likely to rank highly cited papers better (Beel & Gipp, 2009). Consequently, we assume that the higher the correlation between quality and citations, the better for the community.

2.5. For me, the coolest result is that reference list lengths help and help a lot across the board. It's a bit surprising that reading budgets are not nearly as useful.

Thanks for noting this.

A recent empirical paper by Kozlowski, et al., 2023 may be of interest regarding referencing budgets. They find a decline in the amount of uncited work, likely a result of the trend of longer reference lists. Particularly relevant to our study is that they find a decrease in overall citation inequality, primarily driven by the reduction in uncited works. If works that remain uncited are less often sources of substantive inspiration, then this reduction — coupled with a decrease in inequality of citation distribution that encompasses uncited papers — can largely be ascribed to rhetorical citing (It is also encouraging to see that their empirical results are consistent with our model).

2.6. Were hypotheses 1-4 pre-specified and preregistered, or are they a rhetorical device for the presentation? If the latter, I am uneasy with framing claims as hypotheses unless this is done before the results are known.

Fair point. We rephrased our claims as “research questions” without specifying an expected outcome. Specifically, we now write them as:

Research Question (RQ) 1: How do rhetorical citations impact the citation-quality correlation?

RQ 2: How does rhetorical citing impact the citation churn?

RQ 3: How does rhetorical citing impact citation inequality?

RQ 4: How does the literature size impact the community health metrics?

RQ 5A: How does the citing budget impact the community health metrics?

RQ 5B: How does the reading budget impact the community health metrics?

We also removed all sentences “These results support Hypothesis XXX” in the result section.

2.7. My biggest problem with the paper is that I simply don't find it convincing as an argument about what happens in the real world. This is admittedly stylistic preference about models but I find that the present one lands in an uncanny valley between simple analytic models that reveal relationships that are almost unavoidable, and highly detailed models that try to encapsulate much or all of what is known of real-world behavior. Instead we get a complicated agent-based model that has too many moving parts to be able to easily assess external validity — and here I mean not just to the parameter choices in the model, but to the more fundamental decisions that the authors made in designing a model of citing behavior. As a result I am convinced that agents who behave as the agents in this model behave will exhibit the comparative patterns described in the paper. But I'm not convinced that these patterns hold for actual researchers' actual citation practices, and I don't know what basic findings to take away about those actual citation practices. And thus I'm not sure how valuable the paper is in the end.

We understand these are very reasonable concerns to bring to agent-based modeling papers and, of course. We offer an empirical point, and one proposal for how to think about this paper's contribution.

First, it may be worth noting that this project was inspired in the first place by our reading of the literature on reading and citing practices for a decade if not more, and our dissatisfaction with how un-empirical many citation modeling papers were. Consequently, we wanted to make this model much closer to reality (which helps explain the many moving parts). Some empirical work that appeared after we wrote the initial version of the paper provides further strong support for several key ideas:

- Hoppe et al., 2023 proposed a clever way to identify substantive vs. rhetorical citations – citations added during peer review are very likely to be rhetorical. They find that in the biomedical literature, 11.6% of references are inserted during the peer review stage, and these rhetorical references are much more likely to be to lower-citation papers than the ones that remained throughout the project. In addition to the large-scale survey of (Teplitskiy et al. 2022), this helps support our modeling assumptions that referencing proceeds in stages, with rhetorical references being to lower status papers.
- Kozlowski, et al., 2023, as we mentioned before, is also an empirical piece consistent with our claims.

So we believe the model is based on the best available empirical evidence, although such evidence is of course incomplete and multiple judgment calls regarding the priority of mechanisms had to be made.

Lastly, we humbly suggest one way to view the contribution for those who doubt the external validity of the modeling assumptions: it raises the possibility that rhetorical citations have some positive effects. To our knowledge, this possibility is not recognized by the literature much if at all.

New References:

- Beel, J., & Gipp, B. (2009). Google Scholar's ranking algorithm: The impact of citation counts (an empirical study). In *2009 Third International Conference on Research Challenges in Information Science* (pp. 439-446). IEEE.
- Hoppe, T. A., Arabi, S., & Hutchins, B. I. (2023). Predicting substantive biomedical citations without full text. *Proceedings of the National Academy of Sciences*, *120*(30), e2213697120.
- Kozlowski, D., Andersen, J. P., & Larivière, V. (2023). Uncited articles and their effect on the concentration of citations. *arXiv preprint arXiv:2306.09911*.

Reviewer 3

3.1. This is an incredibly interesting and completely original piece of research that is likely to influence and advance the discussion of citation theories and the relation between citation counts and research quality. The analyses are robust, and the relationship between the stated hypotheses, the proposed model, and the reported results is high. As such, I have no substantial comments on what is IN the paper, this is an exceptional piece of research.

We appreciate your kind comments!

3.2. However, there are topics left out that could deserve some additional consideration from the authors. It may not require inclusion in the analysis but should be considered during the initial framing and the discussion of the results. This is the concept of high-level types of citations and the social processes of citing. As for the types of citations, consider, e.g., the recent paper by Leahey, Lee and Funk (<https://journals.sagepub.com/doi/abs/10.1177/00031224231168074>). While this paper focuses on novelty, the concepts of theory, methods, and results citations are distinct and very present in how articles are cited. Can these types of citations be held equal in discussing inherent paper quality? Are the ICML ratings of theory, methodology, and results papers comparable? This is more of an example of the underlying problem, namely, which kind of quality the ICML ratings measure. This should be discussed further.

Good points. We made the following revisions in response:

Pg. 3: Citations are widely used in science to measure the impact of papers and researchers. The assumption underlying this evaluative use of citations is that when writing papers researchers generally cite prior work to acknowledge intellectual debts (Baldi, 1998; Zuckerman, 1987). The debts can be of different types, including methodological, theoretical, and empirical (Leahey et al., 2023).

Pg. 14: For simplicity, we do not consider different types of underlying quality, such as methodological or theoretical novelty (Leahey et al., 2023; Guetzkow et al., 2004).

Pg. 16: Each submitted paper undergoes a two-stage review process, with two reviewers evaluating it at each stage on originality, technical soundness, clarity, significance, and relevance to the literature. The ratings averaged across a paper's four reviewers follow an approximately normal distribution. No papers received maximum points or 0. Assuming that only papers that score in the top 20-30% are accepted (published) leads to a long-tail distribution of ratings of published papers.

Pg. 26: we model the underlying quality of a paper with a single constant, although recent evidence suggests that different types of quality have distinct citation patterns (Leahey et al., 2023), which merits further exploration in future studies.

3.3. The authors address one type of social process for citing behavior: social influence increases adoption. However, their model overlooks other substantial processes. Namely, authors tend to cite people they are familiar with more (cronyism, in the extreme case, higher discoverability in the more lenient interpretation). One could argue that this influences the reading budget of authors and very clearly which articles are in the pool of papers that can go into the reading budget. This is related to the information-seeking aspect of the paper - do all researchers perform correct, systematic literature searches for their research? This is not what the literature suggests, although there may be field differences in this as well. The ever-increasing amount of available articles to read and cite increases the potential for other signals of virtue, i.e., social markers, as argued by Merton. This entire concept needs to be elaborated.

Thanks. We made the following revisions in response:

Pg. 7: Researchers may also use other criteria to select what to read, such as familiarity with the authors and recommendations (Milard & Tanguy, 2018), although how frequently such criteria drive decisions is debated (Murray & Poolman, 1982). We do not model these factors in the foregoing model, but note that this is a simplification of empirical practices.

Pg 25-26: Third, it is important to acknowledge that our models primarily focus on capturing the dynamics of reading and citing within the short-to-medium term. Over a longer period, we should consider additional dynamics such as the scientific obsolescence of older works, the diminishing weight of previous citations, and the introduction of new research. As we look towards future work, it would be intriguing to incorporate more variability into the reading process. Social markers such as citation counts and journal impact factors may rise in prominence over time and they will in turn influence authors' allocation of reading and selection of papers to read and cite. Intriguing extensions of the model also include the impact of real-world social relationships on the reading and citing processes. For instance, authors might be more inclined to read and cite papers written by their close social connections. Authors may group papers into "tiers," choosing to read everything they can find if it is in their topic and discipline (the top tier), reading more selectively when it is in the same topic but a different discipline (the next layer), and so on. Additionally, we model the underlying quality of a paper with a single constant, although recent evidence suggests that different types of quality have distinct citation patterns (Leahey et al., 2023), which merits further exploration in future studies.

3.4. And it also brings me to my only (minor) annoyance with the paper: at one point, the authors state that this paper proposes a synthesis of existing citation theories. I strongly suggest omitting this statement, as it oversells the paper; The statement seems to suggest that the research is a new theory, which is overreaching. It is, however, a highly interesting and important part of the puzzle in developing such a theory.

Fair point, and we now see it was a stretch. We revised the text as follows:

Pg. 5.: Formulating a comprehensive theory of citing has been a challenge in information and library science for many decades (Cronin, 1981; Bornmann & Daniel, 2008; Tahamtan & Bornmann, 2019). The main existing theories – normative and social constructivist – have been criticized as incomplete as stand-alone theories (Cozzens, 1989; Nicolaisen, 2007). Recent scholarship focuses on synthesizing the theories, for example the "social systems citation theory" of Tahamtan & Bornmann (2022). Our model contributes towards these efforts by integrating diverse citing motivations.

New References:

- Guetzkow, J., Lamont, M., & Mallard, G. (2004). What is originality in the humanities and the social sciences?. *American Sociological Review*, 69(2), 190-212.
- Leahey, E., Lee, J., & Funk, R. J. (2023). What types of novelty are most disruptive?. *American Sociological Review*, 88(3), 562-597.
- Milard, B., & Tanguy, L. (2018). Citations in scientific texts: Do social relations matter?. *Journal of the Association for Information Science and Technology*, 69(11), 1380-1395.
- Murray, S. O., & Poolman, R. C. (1982). Strong ties and scientific literature. *Social Networks*, 4(3), 225-232.
- Tahamtan, I., & Bornmann, L. (2022). The social systems citation theory (SSCT): A proposal to use the social systems theory for conceptualizing publications and their citations links. *Profesional de la Información*, 31(4).

Reviewers' Comments:

Reviewer #1:

Remarks to the Author:

The authors addressed the reviewers' comments appropriately. I recommend to publish the paper.

Reviewer #2:

Remarks to the Author:

The authors have engaged with my comments from the first round in good faith and with considerable effort. I find their arguments compelling, I appreciate the changes they have made, and I personally think the paper is better for it.

I take their point about the role of agent-based models such as theirs, and in any case I don't think my personal sense of aesthetics should stand in the way of publication.

Thus I can enthusiastically endorse this paper for publication.

I did not know that Hoppe paper; it is a nice complement to the present work.

A minor comment: Here, as often, Simkin and Roychowdhury (2005) are cited as evidence that authors don't read the papers they cite. While I don't doubt the conclusion, the evidence has always struck me as unconvincing because they failed to consider an alternative explanation that in my view is more likely: even for papers they have read, authors copy the citation itself from other papers and/or online databases, rather than typing them in anew. This leads to the same patterns the authors claim are evidence that people are citing work they have not read.

Reviewer #3:

Remarks to the Author:

My comments have been adequately addressed, and I am also happy with revisions made due to comments from other reviewers. I recommend publication.

Reviewer 1:

We appreciate the comments. They have been instrumental in refining our work!

Reviewer 2:

We appreciate the comments. We have removed the contentious reference, Simkin and Roychowdhury (2005).

Reviewer 3:

We appreciate the comments. They have been instrumental in refining our work!

Reviewers' Comments:

Reviewer #1:

None

Reviewer #2:

Remarks to the Author:

As I said in my previous review, "I can enthusiastically endorse this paper for publication."

I honestly don't understand why this came back for re-review for a third round of review. All three reviewers concisely approved it for publication in the previous round. What more are the editorial staff looking for?